# Exploring the Pivotal Components Influencing the Side Effects Induced by an Analgesic-Antitumor Peptide from Scorpion Venom on Human Voltage-Gated Sodium Channels 1.4 and 1.5 through Computational Simulation

**DOI:** 10.3390/toxins15010033

**Published:** 2022-12-31

**Authors:** Fan Zhao, Liangyi Fang, Qi Wang, Qi Ye, Yanan He, Weizhuo Xu, Yongbo Song

**Affiliations:** 1School of Life Science and Biopharmaceutics, Shenyang Pharmaceutical University, 103 Wenhua Road, Shenyang 110016, China; 2Faculty of Functional Food and Wine, Shenyang Pharmaceutical University, 103 Wenhua Road, Shenyang 110016, China

**Keywords:** voltage-gated sodium channel, Na_v_1.4, Na_v_1.5, analgesic-antitumor peptide, subtype selectivity, adverse drug reaction, molecular dynamics

## Abstract

Voltage-gated sodium channels (VGSCs, or Na_v_) are important determinants of action potential generation and propagation. Efforts are underway to develop medicines targeting different channel subtypes for the treatment of related channelopathies. However, a high degree of conservation across its nine subtypes could lead to the off-target adverse effects on skeletal and cardiac muscles due to acting on primary skeletal muscle sodium channel Na_v_1.4 and cardiac muscle sodium channel Na_v_1.5, respectively. For a long evolutionary process, some peptide toxins from venoms have been found to be highly potent yet selective on ion channel subtypes and, therefore, hold the promising potential to be developed into therapeutic agents. In this research, all-atom molecular dynamic methods were used to elucidate the selective mechanisms of an analgesic-antitumor β-scorpion toxin (AGAP) with human Na_v_1.4 and Na_v_1.5 in order to unravel the primary reason for the production of its adverse reactions on the skeletal and cardiac muscles. Our results suggest that the rational distribution of residues with ring structures near position 38 and positive residues in the C-terminal on AGAP are critical factors to ensure its analgesic efficacy. Moreover, the substitution for residues with benzene is beneficial to reduce its side effects.

## 1. Introduction

Voltage-gated sodium channels (VGSCs) play important roles in membrane excitability transduction [1]. Mammals express nine subtypes of VGSCs (Na_v_1.1–1.9) according to their different tissue distributions and functions. Modification of each subtype yields different biological responses [2,3]. Among these VGSCs, Na_v_1.4 is mainly responsible for generating action potentials in skeletal muscles. When this subtype is activated, the action potential rapidly conveys excitement through the skeletal muscle fibers and regulates the release of Ca^2+^ from myofibrils to drive the contraction and relaxation of skeletal muscle. Clinical and electromyographical features reveal that mutations in the encoding gene *SCN4A* of Na_v_1.4 can trigger hyperkalemic periodic paralysis and paramyotonia congenital, while sodium channel myotonias may produce gain-of-function changes [4], but loss-of-function mutations may induce hypokalamic periodic paralysis and myasthenic weakness [5]. Encoded by the gene *SCN5A*, Na_v_1.5 is the primary segment within the intercalated disks in atrial and ventricular myocytes [6]. Gain-of-function mutations in this gene are related to the disruption of fast inactivation, persistent sodium current generation, and ventricular action potential prolongation [7]. However, loss-of-function mutations in SCN5A disrupt the membrane trafficking of the channel protein. In addition, the majority of patients carrying these mutations are diagnosed with cardiac diseases such as LQT3, Brugada syndrome, and sick sinus syndrome [8,9].

In previous comprehensive studies, by inhibiting human Na_v_1.7 (hNa_v_1.7) with potential analgesic effects, a kind of β-Scorpion Toxin (β-ScTx) was found and named analgesic-antitumor peptide (AGAP, Figure 1) [10,11]. Based on the genetic evidence that gain-of-function and loss-of-function mutations of this sodium channel coding gene may cause painful syndromes and pain insensitivity, Na_v_1.7 has emerged as a promising and well-validated pain target [12,13,14,15]. Therefore, it is reasonable to believe that this VGSC subtype, which is distributed primarily in the peripheral nervous system, is an ideal target for developing non-addictive therapeutics for pain. However, it is alarming that the off-target effects of the analgesic on Na_v_1.4 and Na_v_1.5 could contribute to the possible side effects on the skeletal and cardiac muscles, which may be due to the relatively high conservation of different VGSC subtypes. Unrestricted VGSC blockage could cause heart failure, paralysis, and respiratory failure because it impairs the activity of Na_v_1.4 and Na_v_1.5, which are the primary sodium channels in the skeletal and cardiac muscles, respectively. In fact, a part of analgesic targeting hNa_v_1.7 has to be abandoned in its clinical trials for the above reasons [16,17,18].

The usage of animal venom to develop novel medicines and serve as pharmacological instruments for monitoring voltage-gated sodium channel function has long been recognized. Among them, the scorpion toxins acting on VGSC can be divided into α- and β-Scorpion Toxins (ScTx) based on their differences in electrophysiological properties and binding sites. The α-ScTx binding on site 3 in VGSC is believed to trap the S4 segment on DIV in an inward position, which prevents it from moving normally in response to depolarization and prolongs action potentials. Meanwhile, β-ScTx can reduce the amplitude of the peak currents and potentiate the activation of the Na^+^ channels by binding to site 4 in VGSCs (Figure 2) [19,20]. For instance, Huang et al. found that an α-like scorpion toxin, OD-1, had an agonistic effect on VGSC and served as a new excitotoxicity and seizure model to explore the underlying mechanism of a novel third-generation antiepileptic drug [21,22]. Unlike that, AGAP from Buthus martensii Karsch (Bmk) is a 66-amino acid neurotoxin that belongs to β-ScTx [23]. The outcomes of our earlier electrophysiological studies on AGAP were exactly consistent with this characteristic. In the whole-cell patch clamp tests on different VGSC subtypes, compared to the control, the peak current was significantly suppressed. The analysis of the current–voltage relationship displayed a negative shift in the voltage dependence of activation following priming depolarizations after exposure to 100 nM AGAP. In contrast, no significant action on V_1/2_ values of inactivation was observed, and the duration of the recovery process stayed mainly unchanged. As a result, this peptide is defined as a kind of β-ScTx by this current–voltage relationship for AGAP-modified sodium currents [24]. Four homologous domains (DI–DIV) form the α-subunit of VGSC (Figure 2). Site 4 is formed by the extracellular loops connecting transmembrane helices S1–S2 and S3–S4 at DII. Likewise, the extracellular loops S1–S2 and S3–S4 are also the principal components of site 3, but this binding site is located on DIV [25]. After activation by a strong depolarization, i.e., the toxin trapped into the VGSC active conformation, the residues of S4 on the voltage sensing domain (VSD) of DII targeted by β-ScTx will be exposed [26]. Cumulatively, subsequent activation and repetitive action potential firing will be enhanced [27]. Therefore, β-ScTx can alert the action potential of VGSCs in the activated state. Extensive experimental and computational studies have provided further insights into the interaction mechanism between β-ScTx and VGSCs. These results identified that substitution of E779 and P782 at DII/S1-S2 or A841, N842, V843, E844, G845, and L846 at DII/S3-S4 reduced the bioactivity of Css4 (a kind of β-ScTx from Centruroides suffusus) on rNa_v_1.2. Furthermore, the replacement of DII in this channel with the counterpart DmNa_v_1 domain from Drosophila melanogaster inverted the insensitivity of AahIT (a kind of β-ScTx from Androctonus australis) to rNa_v_1.2 [28,29].

With IC_50_ values of 3.25 × 10^−8^ M, 2.19 × 10^−8^ M, and 1.41 × 10^−8^ M, respectively, the similar biological activity of AGAP to hNa_v_1.7, hNa_v_1.4 and hNa_v_1.5 is the most likely reason for the occurrence of their adverse effects on the skeletal and cardiac muscles. This possibility was confirmed in subsequent animal experiments. The measurement of heart rate, creatine kinase (CK), and lactic dehydrogenase (LDH) in mice after intravenous injection of AGAP verified the acute toxicity of this peptide to cardiac muscle and could not survive more than six days, even at a low dose of AGAP-treated group. It also led to the absence of motor function tests because no eligible mice survived in this group [30]. To address this problem, we conducted multiple cellular and molecular studies, and the results primarily identified the importance of W38 on AGAP in the specificity of different VGSC subtypes and screened two effective mutants (AGAP^W38G^/AGAP^W38F^). The whole-cell clamp patch test and in vivo experiments indicated that there were no significant effects on skeletal and cardiac muscles after intravenously injecting AGAP^W38G^ compared with the saline-treated group [24,31]. To date, there has not been enough evidence to provide a panoramic mechanistic understanding of how AGAP interacts with different VGSC subtypes. In light of our previous results on the binding modes of AGAP and hNa_v_1.7 [32], we herein elucidated the detailed mechanism of AGAP mutants with hNa_v_1.4 and hNa_v_1.5 through dynamic simulations, revealing the reason why the mutation of one single amino acid can bring about the remarkable alteration of subtype selectivity. We believe these findings are not only beneficial to avoid toxicity to the muscles and myocardium but also helpful to promote progress in developing safer and more effective treatments aimed at VGSC subtypes.

## 2. Results

For ease of presentation in this paper, the amino acid residues on VGSCs were indicated by three-letter abbreviations and on peptides by single-letter abbreviations. The other abbreviations were listed in the Appendix A as well (Appendix A).

### 2.1. Differences in 3D Structures of VSD2^hNav1.4^ and VSD2^hNav1.5^ in Comparison with VSD2^hNav1.7^

MD simulations were carried out to clarify the effects of receptor structure differences on the selectivity of AGAP. The RMSDs of VSD2s on hNa_v_1.4, hNa_v_1.5, and hNa_v_1.7 during MD simulations were 0.39 nm, 0.39 nm, and 0.37 nm, respectively, indicating the similar stability of these three systems (Figure 3). As the major binding site for β-ScTx, the extracellular loops connecting DII/S1–S2 and DII/S3–S4 displayed significant primary sequence alignment identity differences with the transmembrane helices (S1–S4) (Figure 4). The different residues in S1–S3 may interact with conserved negatively charged residues in S4 to form different salt bridges (Figure 5). Specifically, R114 and R117 on S4 directly engage E40 on S1 and E98 on S3 of VSD2^hNav1.7^ through salt-bridge interactions so that the distances between these three helices were shortened. Similar trends were observed for R114 and E40 of VSD2^hNav1.5^ but not for VSD2^hNav1.4^. As a result, the binding site of β-ScTxs, the gap between the two extracellular loops on S1–S2 and S3–S4 of VSD2^hNav1.7^, is much more compact than the gap of VSD2^hNav1.4^ and VSD2^hNav1.5^. The discrepancy between the spatial structures of the active pockets may further explain the different bonding strengths of the same types of inhibitors. However, when the IC_50_ values were identified by patch clamp to detect the different subtype selectivity of AGAP, the results were similar [24,31]. It is, therefore, still necessary to further explore the interaction mechanism of AGAP with hNa_v_1.4, hNa_v_1.5, and hNa_v_1.7.

### 2.2. Analysis of the Binding Modes of AGAP and the W38G/W38F Mutant with VSD2^hNav1.4^ and VSD2^hNav1.5^

From 100 ns MD simulations, the static binding poses of AGAP and its two mutants AGAP^W38G/W38F^ with VSD2^hNav1.4^ and VSD2^hNav1.5^ were obtained. Similar to hNa_v_1.7, the major interaction regions were located on the β-turn and C-terminal in the AGAP peptides.

#### 2.2.1. A Structural Model for the β-ScTx-hNa_v_1.4 Complex

Six residues in the β-turn (W38, A39, V41, Y42, G43 and N44) participate in the combination of AGAP with VSD2^hNav1.4^ (Figure 6a). Among these residues, W38, A39, V41, and N44 are well positioned to interact with the bond DII/S1-S2 loop, and V41, Y42, and G43 interact with the DII/S3–S4 loop. Moreover, W38 and V41 have wide ranges of interactions with VSD2^hNav1.4^. Specifically, W38 contacts Met37/His41/Pro43/Leu52 in the DII/S1–S2 loop, while V41 contacts Met39/Thr53 in the DII/S1–S2 loop and Arg111/Arg114 in the DII/S3-S4 loop. Additionally, K62, C63, N64, and G65 in the AGAP C-terminal are also active in binding with VSD2^hNav1.4^. They interact with Leu107/Ser108 in DII/S3–S4 except for K62, which interacts with Glu40 in the DII/S1–S2 loop. Apart from residues in these two regions, Y5, Y14, F15, and Y35 in wild type (WT) contributed to the bindings as well.

When W38 is substituted in WT by G38, the number of residues with direct contact decreases dramatically (Figure 6b). In the wild-type W38 β-turn, all residues bound to VSD2^hNav1.4^ pointed toward the DII/S1–S2 loop. AGAP^W38G^ failed to continue to form multiple interactions with VSD2^hNav1.4^. Meanwhile, V59 and C62 in the AGAP^W38G^ C-terminus, respectively, bind to Tyr42 and Gln105 in VSD2^hNav1.4^. In addition, F15 also contributes significantly to binding by interacting with Arg114.

When W38 is substituted in WT by F38, six residues in the β-turn (Q37, F38, A39, V41, Y42, and N44) participate in the combination of AGAP with VSD2^hNav1.4^ (Figure 6c). Among these residues, Q37, F38, and A39 were positioned to interact with the DII/S1–S2 loop, while V41, Y42, and N44 interacted with the DII/S3–S4 loop. Similar to the WT, V41 in AGAP can directly contact these two extracellular loops. R58 and G61 in the AGAP^W38F^ C-terminus combined with Tyr42 and Arg111, respectively, in VSD2^hNav1.4^. The interactions between D8/C12/F15 with Gln105, Y21 with Pro102, Y35/W47 with Asp49 and Met44 were also important in AGAP binding on VSD2^hNav1.4^.

Evidently, the binding modes of WT and AGAP^W38F^ to hNa_v_1.4 bear more striking resemblances compared to AGAP^W38G^, suggesting that the ring structure at position 38 has a critical effect on its combination with the DII/S3–S4 loop. In contrast, the interactions of the peptides with the DII/S1–S2 loop were more stable. Although the active pocket will naturally converge if the peptide contacts both extracellular loops, the inherent spacious active pocket characteristic of VSD2^hNav1.4^ is destined to affect the affinity of AGAP.

#### 2.2.2. A Structural Model for the β-ScTx-hNa_v_1.5 Complex

Seven residues in the β-turn of WT (Q37, W38, A39, G40, V41, Y42, and N44) interact directly with VSD2^hNav1.5^ (Figure 7a). Most of the residues were positioned to combine with the bound DII/S3–S4 loop except for Q37 and G40. Multiple interactions were observed between W38 and Glu98/Arg111/Arg114 and between N44 and Glu98/Ser102/Met104. Multiple residues in the C-terminal segments participated in the combination of the WT to DII/S3–S4 loop in VSD2^hNav1.5^. Moreover, Y5 and W47 in WT also contributed to the interactions with Asn43 in the DII/S1–S2 loop. On the whole, the ligand was biased to the side of the DII/S3–S4 loop.

When W38 was substituted in WT by G38, the binding of this residue to VSD2^hNav1.5^ was abolished. Unlike VSD2^hNav1.4^, the number of residues in the β-turn that interacted with the receptor was equal to the number of residues in the β-turn that interacted with the receptor in the WT (Figure 7b). In contrast, almost all interactions between the AGAP C-terminus and VSD2^hNav1.5^ disappeared except for the contact between C63 and Asn106. Overall, AGAP^W38G^ binds to the middle of the active pocket, with approximately equal distance to both of the extracellular loops.

When W38 was substituted in WT by F38, the role of this residue in combination with VSD2^hNav1.5^ was retained (Figure 7c). The ligand bonded with the receptor was biased to the side of the DII/S1–S2 loop due to the weakening interaction between AGAP^W38F^ and the DII/S3-S4 loop. The function of the C-terminus is similar to the function of the WT. However, Y5, N19, Y35, and W47 are also important in peptide binding to hNa_v_1.5. Among these residues, Y5 on β-sheet I and W47 on β-sheet II in the VSD are adjacent in space and close to Q37. These three residues may form a signature binding region to hNa_v_1.5 compared to the other two VGSC subtypes.

The comparison of the binding modes of AGAP and the W38G/W38F mutant with hNa_v_1.4, hNa_v_1.5, and hNa_v_1.7 shows that Q37, 38th, G40, V41, Y42, and N44 in the β-turn comprise a crucial binding region of the peptide when it interacts with VGSCs. Among these binding modes, Q37 is preferentially bound by DII/S1-S2 in VSD2^hNav1.5^, while N44 always contributes significantly to the binding of the three VGSC isoforms. The ring structure at position 38 critically affects its combination with the VGSC. Alternatively, the functions of the C-terminus in combinations with hNa_v_1.4 and hNa_v_1.5 are roughly identical and far less powerful than those of the C-terminus in combinations with hNa_v_1.7. Another interesting difference is that C63 in this segment is always bound with Asn106 in VSD2^hNav1.5^ or the residue at position 105 in VSD2^hNav1.4^ and VSD2^hNav1.7^, which seems to indicate that the residues at this position in the different receptors have important implications for the subtype selectivity of the toxins. In addition, F15 and Y5/W47 are also important for the binding of VSD2^hNav1.4^ and VSD2^hNav1.5^, respectively.

### 2.3. Analysis of Dissociation Pathways of the AGAP/AGAP^W38G^/^W38F^ Mutant with VSD2^hNav1.4^ and VSD2^hNav1.5^ by SMD Simulations and PMF Calculations

#### 2.3.1. Differences in Conformations of AGAP and the W38G/W38F Mutant with VSD2^hNav1.4^ and VSD2^hNav1.5^

Based on the SMD method, the specific modes and interactions of critical residues are depicted precisely by analyzing the dissociation of the peptides from the three VGSC isoforms. The results show that AGAP, AGAP^W38G,^ and AGAP^W38F^ are separated completely from VSD2^hNav1.4^ after 3680 ps, 2950 ps, and 3360 ps (Figure 8a), as is VSD2^hNav1.5^ after 2880 ps, 2920 ps, and 3070 ps (Figure 8b). PMF indicates that the binding free energy of AGAP, AGAP^W38G,^ and AGAP^W38F^ are 190.64 kJ·mol^−1^, 175.68 kJ·mol^−1,^ and 150.53 kJ·mol^−1^ to hNa_v_1.4 (Figure 9a), as well as 164.81 kJ·mol^−1^, 146.19 kJ·mol^−1,^ and 149.48 kJ·mol^−1^ to hNa_v_1.5 (Figure 9b). Apparently, WT has a higher affinity for hNa_v_1.4 and hNa_v_1.5 than the two mutants. Moreover, the same change trend is expressed between dissociation time and binding affinity. Overall, the simulation and previous patch clamp experimental results are consistent with each other. Furthermore, the proportion of electrostatic and VDW interactions in AGAP binding with hNa_v_1.5 were found not as regular as it was with hNa_v_1.4 and hNa_v_1.7, which were dominated by only one single type of interaction (see Section 2.3.2 and Section 2.3.3 for details).

According to the comparison between the conformations of the peptides about dissociation from the receptors (Figure 10) with binding modes (Figure 6 and Figure 7), the interactions of W38 in the β-turn with DII/S1-S2 and K62 in the C-terminal with negatively charged residues in DII/S4 contribute significantly to the combination of WT to hNa_v_1.4 and hNa_v_1.7, but not to hNa_v_1.5. However, mutations of position 38 can partially decrease the direct connection between G38/F38 and DII/S1–S2 but completely destroy it between the C-terminal and II/S4. Moreover, a broad binding region of β-turns to the three VGSC subtypes ensures stable and strong contact between the toxins and receptors. The interactions of the more flexible C-terminal to the VGSCs are easily reformed.

#### 2.3.2. Specific Types of Interactions of Important Residues in the β-ScTx-hNa_v_1.4 Complex

The decompositions of the binding free energy of ligand–residue pair interactions are employed to investigate how critical components influence the affinity and selectivity of the peptides of different VGSC subtypes. The calculation results show that the average energy contributions in van der Waals (VDW) during the dissociations of AGAP, AGAP^W38G,^ and AGAP^W38F^ from VSD2^h^Na_v_^1.4^ are −215.77 kJ·mol^−1^, −135.54 kJ·mol^−1,^ and -187.15 kJ·mol^−1^, respectively, whereas in electrostatic interactions, they are −185.63 kJ·mol^−1^, −100.74 kJ·mol^−1,^ and −178.81 kJ·mol^−1^, respectively. It follows that VDWs bear greater responsibility than electrostatic interactions to the combinations. In contrast, our previous study indicated that the latter interaction type is the dominant factor leading to the differences in the binding free energy of the three peptides to VSD2^hNav1.7^.

In particular, four residues (W/G/F38, A39, N42, and N44) in the β-turn play vital roles in VSD2^hNav1.4^ trapping (Table 1, Appendix A and Appendix A). In accordance with hNa_v_1.7, substitution W38G strongly diminished the toxin binding affinity due to steric hindrance and H-bond repulsion between this residue and DII/S1–S2. Y42 in WT accepts a π-cation contact from Arg111 in DII/S4, which in mutants is H-bonded with DII/S3–S4 for identical contribution. N44 in WT forms weak H-bonds with the two extracellular loops, which are strong with one loop in mutants. A residue located on the loop between α-helix and β-sheet I, F15, also significantly contributes by forming a π-cation contact with the highly conserved Arg114 in DII/S4. Interestingly, the important function of F15 is to work only when the toxins are bound with hNa_v_1.4. Therefore, we deduced that this particularity is attributed to the discrepancy in the 3D structures of the three VGSC subtypes. Additionally, unlike hNa_v_1.7, residues in the C-terminus lack powerful interactions with hNa_v_1.4, although they are involved in the combination of the receptor and toxins.

#### 2.3.3. Specific Types of Interactions of Important Residues in the β-ScTx-hNa_v_1.5 Complex

The calculation results show that the average energy contributions in van der Waals (VDW) during the dissociations of AGAP, AGAP^W38G,^ and AGAP^W38F^ from VSD2^hNav1.5^ are −169.03 kJ·mol^−1^, −167.40 kJ·mol^−1,^ and −218.49 kJ·mol^−1^, respectively, whereas in electrostatic interactions, they are −267.36 kJ·mol^−1^, −158.96 kJ·mol^−1,^ and −174.17 kJ·mol^−1^, respectively.

Specifically, seven residues (Q37, A39, G40, V41, N42, G43, and N44) in the β-turn have significant contributions to electrostatic interactions in all complexes (Table 2, Appendix A and Appendix A). Of these, N44 can always accept connection with DII/S3-S4. Similar to hNa_v_1.4 and hNa_v_1.7, the ring structure at position 38 still has a critical effect on the combination with hNa_v_1.5. Notably, Y5 and W47, which are adjacent to each other and Q37 in space, always keep in direct contact with Asn43 in DII/S1–S2. Moreover, the residues in the DII/S1-S2 contact with Q37 in the peptide are also close to Asn43. Therefore, we inferred that the interaction surface formed by Y5, Q37, and W47 reacts with the new characteristics of the binding pose of the toxins with hNa_v_1.5. Further analysis reveals that the production of the feature residues is derived from the opposite distribution of the hydrophobicity of the extracellular loop in DII/S1-S2 on hNa_v_1.5 and hNa_v_1.4/hNa_v_1.7 (Figure 11). Because it is located in the middle of this loop on VSD2^hNav1.5^, hydrophilic Asn43 is able to directly contact Y5 and W47 on AGAP through electrostatic interactions. Likewise, the roles of residues in the C-terminus are limited in the combination of VSD2^hNav1. 5^ and toxins. Although the negatively charged R58 in this region can contact the positively charged Asn43 in DII/S1–S2, the contribution is small due to the remote distance between them.

**Table 2 toxins-15-00033-t002:** Average total-residue interaction of AGAP and its mutants with VSD2s on hNa_v_1.5 during the dissociation process.

Region	Residue	Total Interaction (kJ·mol^−1^)		
		AGAP	AGAP^W38G^	AGAP^W38F^
β-sheet I	Y5 * ^1^	−17.07	−15.63	−28.23
loop between α-helix and β-sheet I	R18	-	−2.83	-
	N19	-	-	−35.77
β-sheet III	Y35	−11.30	-	−48.22
	C36	-	-	−8.85
β-turn	Q37 *	−37.77	−44.46	−38.32
	W38	−83.59	-	-
	G38	-	-	-
	F38	-	-	−29.71
	A39 *	−10.93	−10.09	−11.77
	G40 *	−25.21	−31.28	−29.44
	V41 *	−16.80	−36.36	−25.75
	Y42 *	−23.21	−38.87	−16.02
	N44 *	−64.25	−24.25	−45.47
β-sheet II	W47 *	−31.40	−23.19	−19.55
C-terminal	R58	−19.23	-	−8.82
	V59	−5.30	−2.39	−2.30
	K62	−33.93	-	-
	C63	−21.27	−14.85	−1.11

^1^ The characters in “*” played important roles in all three systems.

**Figure 11 toxins-15-00033-f011:**
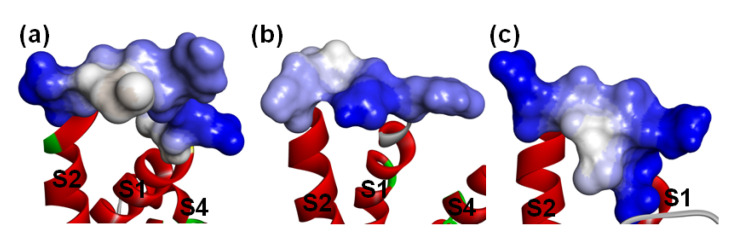
Hydrophobicity of the loop between S1-S2 on VSD2s on VGSCs. (**a**) hNa_v_1.4; (**b**) hNa_v_1.5; (**c**) hNa_v_1.7. The default color spectrum used is blue–white–brown. Surfaces in blue correspond to hydrophilic residues, whereas surfaces in brown correspond to hydrophobic residues. Residues at positions 40–46 on VSD2s are shown on the surface.

## 3. Discussion

Combined with our previous study [32] and this research, two major factors are highly related to the selectivity of AGAP and its mutants to hNa_v_1.4, hNa_v_1.5, and hNa_v_1.7. First, there are progressive dissimilarities in the 3D structures of AGAP bound, in which VSD2^hNav1.7^ is the narrowest, VSD2^hNav1.5^ is wider, and VSD2^hNav1.4^ is the widest. These differences in the binding poses established the feature residues of AGAP binding. Second, the affinity of WT AGAP to VGSCs is always higher than the affinity of the two mutants, according to the binding energy calculated by simulations and the IC_50_ values detected by experiments. This evidence fully demonstrates that the residue at position 38 on AGAP is one of the dominant factors affecting the selectivity of AGAP to the three VGSC subtypes. Moreover, the results indicate that the ring structure of this position is presented as the hinge structure, which provides a significant contribution when it contacts the channels through VDW interactions. Similarly, K62 is also the pivotal residue in the AGAP C-terminal, which may offer an impressive energy contribution to the negatively charged residue on VGSCs when salt bridges form. Benefiting from the narrow active pocket of AGAP binding to VSD2^hNav1.7^, K62 can contact negatively charged Asp103 and Glu105 in the DII/S3-S4 loop through powerful electrostatic interactions.

Additionally, a significant correlation was found between W38 and K62. For intermolecular interactions, these two residues displayed a synergistic effect to determine the selectivity of AGAP to different VGSC subtypes. For instance, although the affinity of W38 and F38 is similar to the affinity of hNa_v_1.4, the binding free energy of WT AGAP is much higher than that of AGAP^W38F^ because of the formation of a salt bridge between K62 on WT and Glu40 on VSD2^hNav1.4^. In contrast to AGAP^W38F^, the contribution of position 38 on AGAP^W38G^ is minimal, while the contribution of K62 is still considerable when the toxin contacts VSD2^hNav1.7^. Finally, the affinities of the two mutants to hNa_v_1.7 are equivalent. For intramolecular interactions, the function of K62 is affected by the substitution of residue at position 38 as well. The electrostatic interactions between K62/C63 and the negatively charged residues in the DII/S3–S4 loop on VSD2^hNav1.7^ were discovered to arise from the existence of the internal reaction chain on WT (C63-C12-D8-N11-R58-Y42), which limits the swing of the C-terminus. On AGAP^W38F^, this chain is broken because of the abolition of the interaction between Y42 and R58 by A39. In WT and AGAP^W38G^, the group of A39 interacting with Y42 is occupied by the combination with the receptor. Likewise, the formation of the salt bridge between K62 and the negatively charged residue in the DII/S1-S2 loop on VSD2^hNav1.4^ comes from a similar internal chain on WT (C63-C12-D8-N11-G61-Y42). After the mutation, the electrostatic interaction between Y42 and G61 is distributed because the reaction group in the former is occupied by A13 and Y14. The reason is the swerve of the benzene on Y42 because of the interactions between it and Gln105/Arg111 on VSD2^hNav1.4^. In the meantime, the shortage of the internal reaction chain to restrict the direction of the C-terminal on AGAP results in the modest energy contribution of this region when it is bound with hNa_v_1.5.

However, there is still a sensitive difference in the affinity of the residue at position 38 to hNa_v_1.5 on WT, AGAP^W38G,^ and AGAP^W38F^. This result highlights that a single factor is not enough to determine the selectivity of AGAP to the three VGSC subtypes.

Additionally, some limitations of this study should not be omitted. Firstly, following the change of membrane potential, there are three different statuses: open, inactive, and closed state. Herein, the structures in an open state were selected to explore the interactions between AGAP and different VGSC subtypes. However, the combination of toxins and sodium channels should be dynamically regulated. For β-ScTx, a voltage sensor trapping model is universally accepted. In this model, the toxin first attaches to its receptor site on VGSC in its inactive state, leading to a concentration-dependent decrease of the peak current. As the channel is activated by a strong depolarization, a new binding site on the channel to β-ScTx is exposed due to the change of conformation of VGSC. Finally, the tightly bound toxin traps the activated conformation of the sodium channel in a process independent of unimolecular concentration [27,33]. This could possibly be the reason for parts of the studies exploring β-ScTx with activated state VGSC through computational and/or experimental approaches [34,35]. For this work, as mentioned above, the outcoming of the patch clamp test manifested that there was little effect of AGAP on the inactivation process [24]. In this way, AGAP appears to suppress the sodium channels, mainly in the open state. Therefore, the interactions between this peptide and activated VGSC were explored in our study. Nevertheless, it does not mean that the complete dynamic process of the toxin binding to the channels is fully considered. These variables should be taken into account in our further studies. Secondly, AGAP draws our attention because of its analgesic effect targeting hNa_v_1.7. Furthermore, its potential skeletal and cardiac muscle toxicity, as well as the similar biological activity to hNa_v_1.7, hNa_v_1.4, and hNa_v_1.5, led us to pay more attention to the interactions between AGAP and these three VGSC subtypes. After intravenous injection of AGAP, central nervous system diseases such as epilepsy and migraine were not observed in mice, so the combinations of AGAP with the VGSC subtypes mainly distributed in this region (Na_v_1.1, Na_v_1.2, Na_v_1.3, and Na_v_1.6) were not examined in this study. On the other hand, Na_v_1.7, Na_v_1.8, and Na_v_1.9 are primarily expressed in the peripheral nervous system, and all have the potential to be a non-addictive analgesic target. Our previous studies by whole-cell clamp patch indicated that AGAP could only suppress the activity of hNa_v_1.8 with a 25% reduction in peak current while hNa_v_1.7 with a reduction in 68% [24]. Limited by the experimental materials, the inhibiting effect of AGAP on hNa_v_1.9 were not discussed in our research. In future investigations, the mechanism of this toxin and other VGSC subtypes should be complemented through both computational simulation and experimental methods. Thirdly, the rational design scheme of AGAP proposed in this paper still needs to be tested with additional experiments. In the sequel of our study, the new mutants will be obtained by genetic engineering. Their actual effects on different VGSC subtypes should be illustrated through whole-cell clamp patch or animal experiments, such as forced swimming test, rota-rod test, and mouse-twisting model.

## 4. Conclusions

Overall, the in-depth simulation analysis revealed several significant commonalities and differences in AGAP binding on the three VGSC subtypes. In general, retaining the ring structure of the amino acid residue at or near position 38, as well as increasing the rational distribution of basic residues in the C-terminal on AGAP, are indicated to be advantageous for improving the affinity of hNa_v_1.7, which was disclosed in our previous studies [32]. In contrast, disruption of the ring structure of F15 and Y5/W47 on AGAP effectively reduced its binding activity for hNa_v_1.4 and hNa_v_1.5.

To elucidate these findings more specifically, from the perspective of AGAP: (i) β-turn is the essential region of AGAP to combine with the VGSC because the majority of the binding residues (at position 37–44) are located within it; (ii) the conserved N44 in the β-turn is always H-bonded to the DII/S3-S4 on the three VGSCs with a prominent energy contribution; (iii) the deficiency of the carbonyl in the R group at position 105 on the DII/S3-S4 loop of hNa_v_1.4 and hNa_v_1.5 may dramatically decrease its electrostatic contacts with Y42 in the β-turn or K62 in the C-terminal on the AGAP; (iv) the unique residues on AGAP may function vitally when combined with different VGSC subtypes, for example, F15 for hNa_v_1.4 and Y5, Q37 and W47 for hNa_v_1.5. From the perspective of three VGSC subtypes: (i) The charged residue at position 49 on VSD2 always accepts a π-cation contact with the residues bearing ring structure (Y35, W38, and Y42) in the β-turn of AGAP; (ii) The highly conserved negatively charged residues in DII/S4 (Arg111 and/or Arg114) participate in the combination with the peptides by forming H-bond or π-cation interactions with G40 and/or V41 in the AGAP β-turns. This response is in accordance with the voltage sensor trapping model, whereby the activated conformation of VSD2, in which DII/S4 moves outward, is trapped by β-ScTxs through strong binding with it. Further research should be undertaken to verify the above results based on experimental methods such as animal toxicity test, western blotting, blood assays, and clamp patch.

We believe that the enlargement of the study of the interaction mechanism of AGAP with hNa_v_1.4, hNa_v_1.5, and hNa_v_1.7 will shed more light on elaborating the selectivity of scorpion toxins to different VGSC subtypes. Furthermore, this research provides more abundant theoretical knowledge and references for developing selective medicines targeting VGSCs.

## 5. Materials and Methods

### 5.1. Homology Modeling and Molecular Docking

The theoretical model of hNa_v_1.4 was obtained from the Protein Data Bank (PDB ID: 6AGF). Five hundred conformations of hNa_v_1.5 and AGAP were built through Modeler 9.9 [36] due to the protein sequences obtained from UniProt (accession numbers were Q15858 and Q95P69, respectively) [37]. Crystal structures of cardiac sodium channel from Rattus norvegicus (PDD ID: 6UZ0) and α-Tx11 from Buthus martensii (PDB: 2KBH) were selected as the templates for homology modeling because of their high sequence identity with hNa_v_1.5 and AGAP, respectively. Models with the least discrete optimized protein energy (DOPE) score were validated by Ramachandran plots and profile-3D and chosen as the best ones. Two mutants of AGAP (AGAP^W38G^/AGAP^W38F^) were obtained by modeler site-directed mutagenesis. To accurately predict the three-dimensional structures of the target sequences in a real physiological environment, molecular dynamics (MD) simulations were applied. The parameters used here are described in Section 5.2.

The optimized structures of AGAP and its mutants were docked into the binding sites on VSD2^hNav1.4^ and VSD2^hNav1.5^ by ZDOCK, which is suitable for studying the interactions between biomacromolecules [38]. The active pockets were restricted in the extracellular region of VSD2. The matching algorithms are used to carry out the process of docking. In accordance with the RMSD cutoff, 2000 poses were divided into 60 clusters with an angular step size of 6°. The small angular step size ensures that the most possible binding modes can be considered in the docking. The cluster containing the largest number of docking poses usually figures out the binding site that the ligand is most likely to combine. Furthermore, the configurations with the lowest binding energy in the largest cluster were screened out for further simulations after excluding those in direct conflict with the position of the membrane and published research results about the binding site of β-ScTx with VGSC [20,28,29]. However, the molecular docking was carried out in a vacuum, which is insufficient to reflect the realistic conformations of AGAP with VGSCs. Therefore, molecular dynamic simulations were needed to optimize these conformations.

### 5.2. Molecular Dynamics

The prediction structures and docking complexes were carried out on 100 ns time scale molecular dynamics simulations utilizing the GROMACS 2018 package [39]. A 1-palmitoyl-2-oleoyl-sn-glycero-3-phosphocholine (POPC) bilayer model was used with a united-atom force field [40] to describe the phospholipid bilayer of hNa_v_1.4 and hNa_v_1.5, and GROMOS-53a6 force-field parameters were assigned to the other parts in the systems. InflateGro methodology [41] was performed to accurately embed these two channels into the lipids. The SPC water model [42] was introduced as the solvation, and counterions were added to neutralize the systems. The steepest descent algorithm and conjugate gradient algorithm were used to minimize the energy and remove the bad contacts first. Subsequently, the simulation conditions were heated to 310 K using a modified Berendsen thermostat [43] with non-hydrogen solute atoms restrained. Simulation in the NPT ensemble follows this desired temperature and 1 atm constant pressure for 1 ns. The equilibration systems were subjected to a 100 ns MD simulation with no constraints applied. Moreover, the particle mesh Ewald (PME) method [44] and LINear Constraint Solver (LINCS) [45] were performed to assess the long-range electrostatic interactions and redress the lengths of all the bonds. The MD trajectory and snapshots were saved every 10 ps and analyzed by the tools of the Gromacs package, PyMol [46], and VMD [47]. The equilibrated trajectory was extracted to perform cluster analysis using the Gromos clustering algorithm with a tolerance of 0.15 nm for root–mean–square deviation (RMSD) [48].

### 5.3. Steered Molecular Dynamics and PMF Calculations

Steered molecular dynamics (SMD) simulations were carried out on the equilibrated model after MD [49]. With a biasing force constant of 500 kJ·mol^−1^·nm^−2^ and a pulling velocity of 0.001 nm·ps^−1^, a force pulled AGAP/AGAP^W38G^/AGAP^W38F^ away from the binding surface of VSD2^hNav1.4^ and VSD2^hNav1.5^ along the z-dimension in the 5 ns simulation runs at 310 K and 1 atm. Herein, with respect to the atoms of VSD2s, the β-ScTxs were taken to be static. Snapshots of the model were captured every 10 ns. The umbrella sampling method, weighted histogram analysis method [50], and PMF (potential of mean force) curves for this process were calculated to describe the binding free energy between the ligand and receptor.

## Figures and Tables

**Figure 1 toxins-15-00033-f001:**
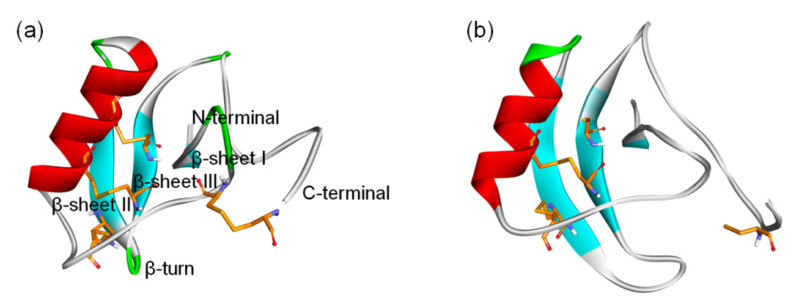
3D structure of β-ScTxs. (**a**) The crystal structure of a toxin from the scorpion Centruroides noxius Hoffmann (2YC1); (**b**) The structure model of AGAP. Residues forming four disulfide bonds are orange; α-helices and β-sheets are colored red and blue, respectively. Naming notations of secondary structures are labeled in (**a**).

**Figure 2 toxins-15-00033-f002:**
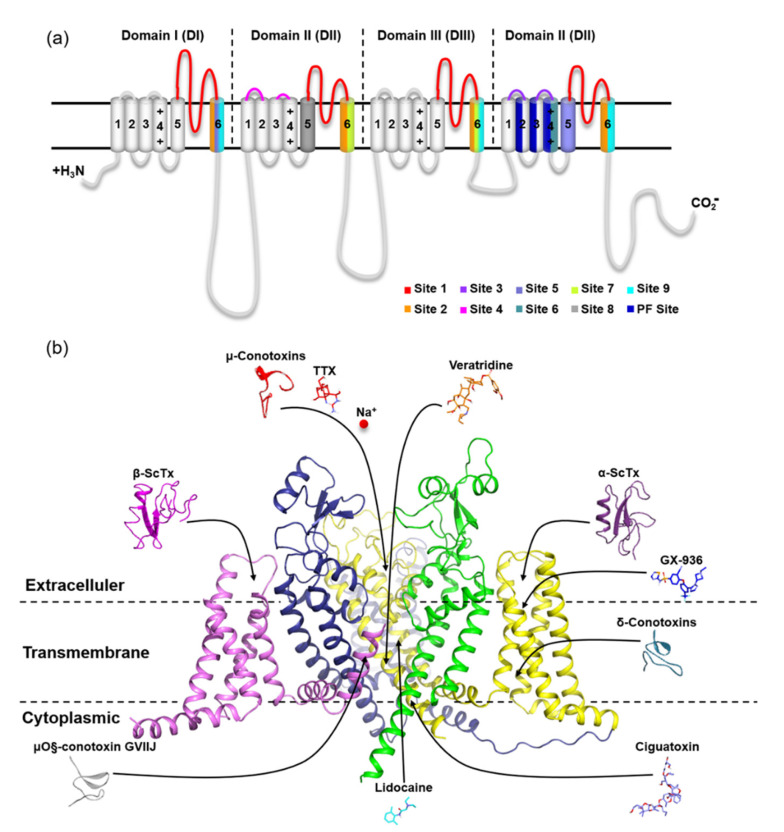
Location of nine receptor sites on mammalian VGSC α subunits. (**a**) Topology of mammalian VGSC α subunits indicating binding sites. Representative inhibitors of sites 1–7 and 9 and the PF site on VGSC are indicated with arrows pointing to the general location of their respective primary receptor sites and colored; (**b**) Side view of the human Na_v_1.7 subunit complex structure (6J8G) in part with the VSD of DI and PM of DII removed for clarity. The representative inhibitors of site 8 (pyrethroids) are synthetic analogs of pyrethrins and are, therefore, not shown here. The PDB codes for peptide toxins are μ-conotoxin (1TCG), α-scorpion toxin (2ASC), β-scorpion toxins (2YC1), δ-conotoxin (1G1P), and μO§-conotoxin GVIIJ (2N8H).

**Figure 3 toxins-15-00033-f003:**
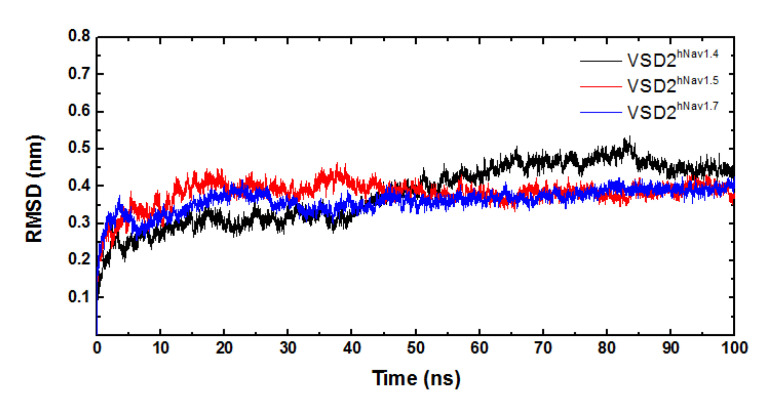
RMSD curves of systems of VSD2s on hNa_v_1.4, hNa_v_1.5, and hNa_v_1.7.

**Figure 4 toxins-15-00033-f004:**
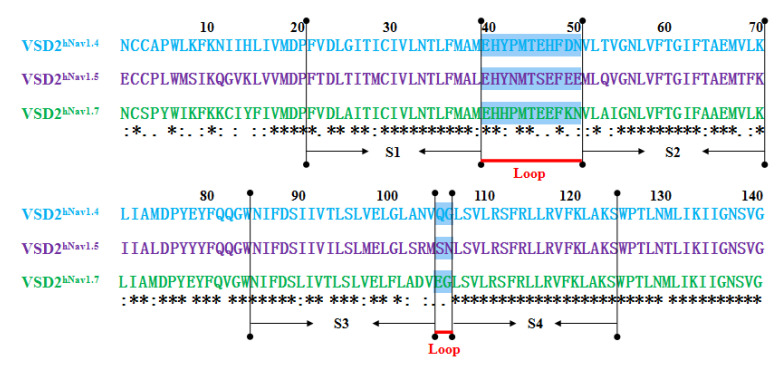
Sequence alignment for VSD2s on hNa_v_1.4, hNa_v_1.5, and hNa_v_1.7. “*” means the residues in this location are identical in three VGSC isoforms; “:” means similar, and “.” means a little similar. Dashed lines in green indicate the extracellular loops connecting S1–S2 and S3–S4 on VSD2s.

**Figure 5 toxins-15-00033-f005:**
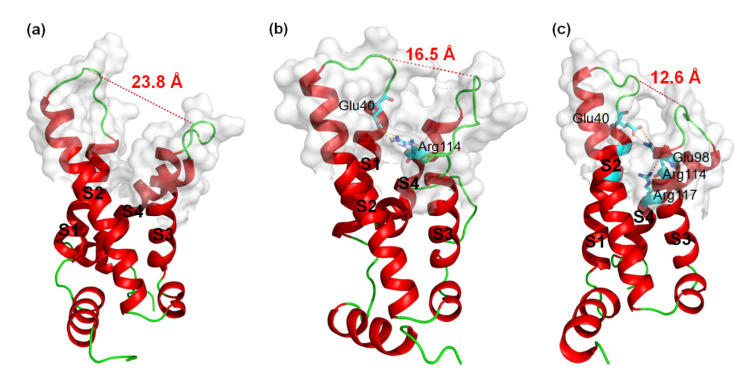
3D structures of VSD2s on hNa_v_1.4, hNa_v_1.5, and hNa_v_1.7. (**a**) VSD2^hNav1.4^; (**b**) VSD2^hNav1.5^; (**c**) VSD2^hNav1.7^. Dashed lines in orange indicate salt bridges; in red are widths of the active pockets formed by loops between DII S1–2 and DII S3–4; amino acid residues formed by salt bridges are in sticks.

**Figure 6 toxins-15-00033-f006:**
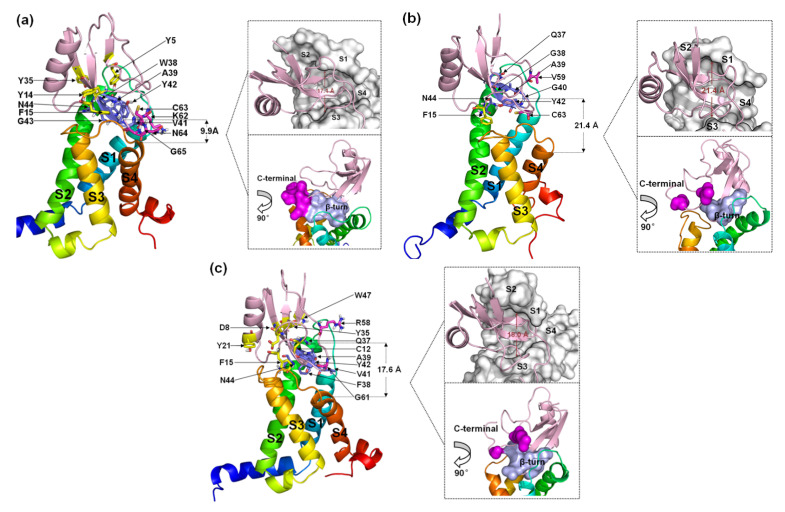
Final binding poses of AGAP (**a**), AGAP^W38G^ (**b**), and AGAP^W38F^ (**c**) with VSD2 on hNa_v_1.4. The interacting residues on the AGAP peptide are represented by sticks: purple (residues 37–44), magenta (residues 58–66), and yellow (all others). The distances between the C-terminus and DII S4 were measured. The width of the active pocket on VSD2^hNav1.4^ is labeled in deep red in the upper insets, which represent the top views of the complex. The interaction surfaces of the β-turn and C-terminal on AGAP/AGAPW38G/AGAPW38F with VSD2^hNav1.4^ are labeled purple and magenta, respectively, in the bottom insets.

**Figure 7 toxins-15-00033-f007:**
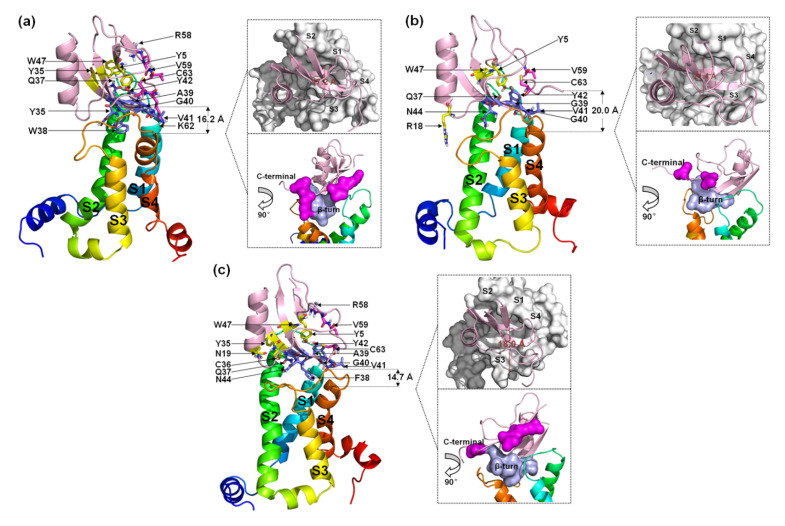
Final binding poses of AGAP (**a**), AGAP^W38G^ (**b**), and AGAP^W38F^ (**c**) with VSD2 on hNa_v_1.5. The interacting residues on the AGAP peptide are represented by sticks: purple (residues 37–44), magenta (residues 58–66), and yellow (all others). The distances between the C-terminus and DII S4 were measured. The width of the active pocket on VSD2hNa_v_1.4 is labeled deep red in the upper insets, which represent the top views of the complex. The interaction surfaces of the β-turn and C-terminal on AGAP/AGAPW38G/AGAPW38F with VSD2^hNav1.4^ are labeled purple and magenta, respectively, in the bottom insets.

**Figure 8 toxins-15-00033-f008:**
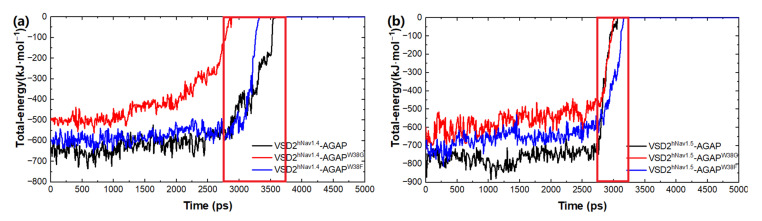
Total energy of AGAP and its mutants with VSD2s on hNa_v_1.4 (**a**) and hNa_v_1.5 (**b**).

**Figure 9 toxins-15-00033-f009:**
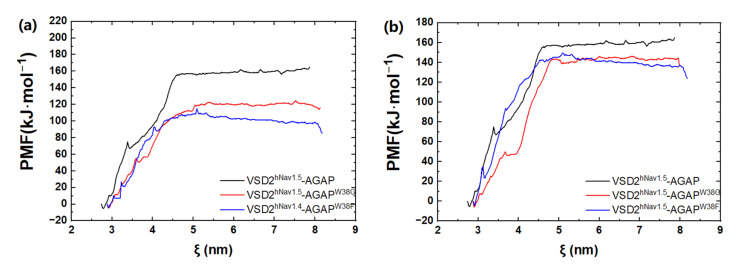
PMF curves of AGAP and its mutants with VSD2s on hNa_v_1.4 (**a**) and hNa_v_1.5 (**b**). ζ represents the reaction coordinate generated by the configurations.

**Figure 10 toxins-15-00033-f010:**
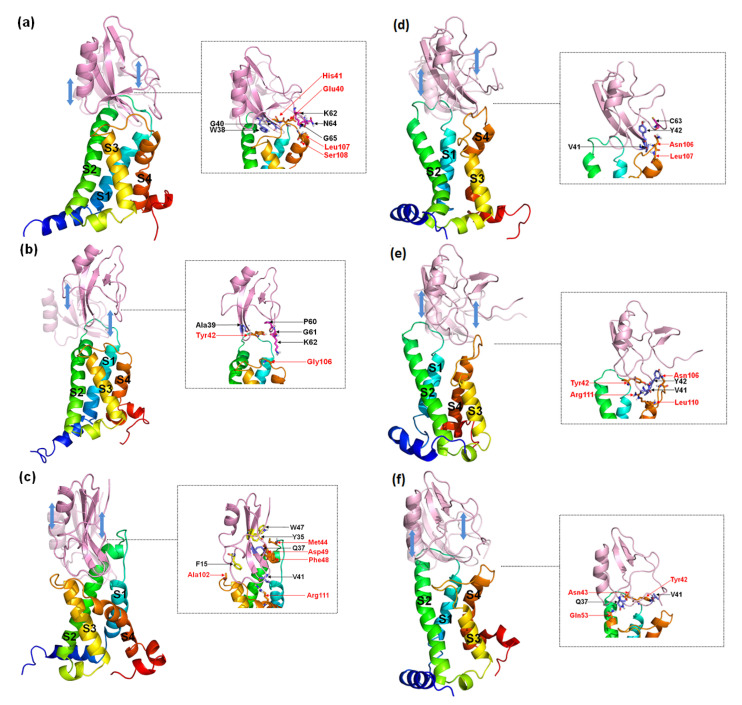
Representative conformations of AGAP (**a**,**d**), AGAP^W38G^ (**b**,**e**), and AGAP^W38F^ (**c**,**f**) with VSD2s on hNa_v_1.4 (**a**–**c**) and hNa_v_1.5 (**d–f**) during the dissociation process. These conformations reflected the state of the receptor and the ligand just prior to complete separation. The ligands in translucency are conformations before pulling. The insets depict the interaction between the receptor and the ligand when they were to be separated. Ligand residues on the β-turn are purple, and those at the C-terminal are magenta; key residues at the interface are marked black. VSD2 hNa_v_1.5 residues are orange and are highlighted in red characters.

**Table 1 toxins-15-00033-t001:** Average total-residue interaction of AGAP and its mutants with VSD2s on hNa_v_1.4 during the dissociation process.

Region	Residue	Total Interaction (kJ·mol^−1^)		
		AGAP	AGAP^W38G^	AGAP^W38F^
β-sheet I	Y5	−0.99	-	-
	D8	-	-	−22.96
loop between α-helix and β-sheet I	C12	-	-	−2.87
	Y14	−2.55	-	-
	F15 * ^1^	−23.10	−20.44	−40.96
α-helix	Y21	-	-	−2.60
β-sheet III	Y35	−3.16	-	−40.40
β-turn	Q37	-	−19.97	−48.43
	W38	−38.99	-	-
	G38	-	−16.60	-
	F38	-	-	−38.59
	A39	−10.16	−12.59	−7.65
	G40	-	−27.39	-
	V41	−37.54	-	−38.77
	Y42 *	−25.09	−18.10	−23.06
	G43	−24.60	-	-
	N44 *	−15.78	−28.25	−24.74
β-sheet II	W47	-	-	−7.25
C-terminal	R58	-	-	−2.75
	V59	-	−2.47	-
	G61	-	-	−14.15
	K62	−68.51	-	-
	C63	−17.33	−10.14	-
	N64	−19.85	-	-
	G65	−25.08	-	-

^1^ The characters in “*” played important roles in all three systems.

## Data Availability

The datasets used and analyzed during the current study are available from the corresponding author upon reasonable request.

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
