# Peer review of "Exploring the Pivotal Components Influencing the Side Effects Induced by an Analgesic-Antitumor Peptide from Scorpion Venom on Human Voltage-Gated Sodium Channels 1.4 and 1.5 through Computational Simulation"

_toxins, 2022, doi:10.3390/toxins15010033_

Round 1

Reviewer 1 Report

Please see attached the file.

Reviewer 2 Report

Dear authors,

I recommend that a minor revision of the manuscript is warranted. I explain my concerns in more detail below and I would ask you to consider each of my comments in your next response.

Minor comments:

Line 11 -  I kindly ask the authors to write the full name of the molecules when they use them in   the text for the first time, not just their abbreviation, eg. NaV1. 4 is a skeletal muscle     voltage-gated sodium channel;

Line 15 -  AGPA must be placed in brackets, otherwise a quick Google search can indicate all kinds of other names with this abbreviation.

Line 38-39  – Please avoid the repetition of gene, second one can be “segment”;

Line 421 - I did not understand which results were published, please indicate the corresponding references here.

Please approach in the conclusion the need of complementing this study with wet lab experimental work to confirm your computational simulation.

Round 2
